# Risk-Based Gastrointestinal Parasite Control in a Tropical Zoological Institute

Yirui Heng * and Delia Hwee Hoon Chua

Mandai Wildlife Group, 80 Mandai Lake Road, Singapore 729826, Singapore; delia.chua@mandai.com
* Correspondence: yirui.heng@mandai.com; Tel.: +65-6360-2231

**Abstract:** The surveillance and treatment of parasites are important features of preventative health-care plans in zoological institutes. The parasite control strategies employed in temperate regions often involve prophylactic anthelmintic treatments during seasons where the burden of gastrointestinal parasites in fecal testing is high. These strategies are, however, not applicable in the tropics, where temperatures remain high throughout the year, allowing continuous parasitic development. A risk-based parasite management strategy was adopted by a tropical zoological institute. For parasite surveillance, routine fecal direct smears and magnesium sulfate flotations were performed to determine parasitic prevalence. The frequency of fecal checks for the year was determined by the frequency at which clinically relevant parasitism (fecal tests that resulted in the animal being treated) was detected during routine fecal checks in the previous year. A yearly anthelmintic drug-class rotation schedule was also implemented. The total number of fecal tests performed per year and the number of animals with clinically significant parasitic disease decreased by 30.0% (637/2126) and 46.9% (207/451), respectively, over the four-year period of the study. Anthelmintic class rotation also improved the efficacy of fenbendazole in treating *Strongyloides* spp. infecting the group of orangutans. This parasite control strategy is a targeted approach to managing preventative healthcare, reducing the work required to perform routine surveillance tests whilst maintaining the health of the collection of animals.

**Keywords:** parasite; anthelmintic; strategy; tropics

## 1. Introduction

Parasite control is of paramount importance to the health of any animal collection. Zoological institutes house many species across various taxa, and a similarly diverse range of parasites infect these animals. As a component of preventive healthcare, routine parasite surveillance and treatment of animals can manage the gastrointestinal parasitic loads in animals and, consequently, the environment. The spatial and temporal incidence of parasites can help to direct husbandry practices to manage the potential spread of infections within a collection, and the changes to parasitic load in an individual animal after treatment can also be an indicator of potential drug resistance within an institute [1].

The routine surveillance of parasites detects the presence of parasites in an individual, but this does not necessarily equate to the animal being diseased. Parasitic disease is accounted for by the interrelationships among the parasites, the host, and the environment. In parasitism, a biological balance is usually reached in the host–parasite relationship, but the influence of the environment or stress factors in zoos (e.g., inclement weather, exhibit renovation) can break this balance, resulting in animals developing disease [1].

Many parasite control strategies have been used by zoological institutes around the world. A review of parasite control programs for captive wild ruminants found that most institutes administered anthelmintics on a regular prophylactic basis (every two to three months) and described how anthelmintic resistance was perceived by many institutes as the main cause of treatment failure [2]. The same survey also found that there were some other

institutes that treated the animals only when indicated in the regularly scheduled fecal examinations [2]. Other studies investigate the efficacy of various parasite control programs in captive wild animals based on variations of season and anthelmintic class, which are similar to the strategies used in the management of parasites in domestic livestock [3–5]. These strategies employ prophylactic anthelmintic treatment of susceptible animals during the warmer seasons, when parasite development occurs and therefore the risk of parasitic infections increases. These strategies are, however, not applicable to zoological institutes in tropical regions, as the constantly high temperatures all year round result in a consistent risk of parasitic infection throughout the year. Biological controls like nematophagous fungi have also been used as feed additives to manage parasitic loads in wild animals under human care, avoiding the use of chemotherapeutic anthelmintics [6].

The parasite surveillance program at Mandai Wildlife Group (MWG) involves routine laboratory analysis of fecal samples obtained from the animals under managed care. When clinically relevant gastrointestinal parasites are detected in the samples, anthelmintic treatments are dispensed by veterinarians. Repeat fecal tests are subsequently performed to assess the efficacy of these treatments. For many years, the veterinary hospital maintained pen-and-paper records of all submitted fecal samples and the test results. In 2019, some of these processes were digitalized, including the log of fecal samples submitted, records of the laboratory test results, and the treatments dispensed. The purpose for this change was for better record keeping of all fecal laboratory analyses performed at the veterinary hospital, and for easier access of these records by veterinary and animal care staff, as the records could now be accessed online anywhere. The improved accessibility of the test results by the animal caretakers improved keeper compliance with the routine fecal tests of the animals under their care and allowed for detailed analysis of the test results obtained.

A new risk-based gastrointestinal parasite surveillance plan was also implemented in the same year. Instead of the previous practice of sampling all animals routinely twice a year regardless of the incidence of parasitism, the frequency of sampling of the different groups of animals was varied based on the parasite treatment frequency of the group of animals in the preceding year. A stricter adherence to an annual anthelmintic rotation schedule was also implemented. The hypothesis was that this targeted strategy of parasite surveillance and management would improve the efficiency of work processes by reducing the number of routine parasite surveillance tests that need to be performed annually, whilst adequately managing the parasitic prevalence and health of the animals.

## 2. Materials and Methods

### 2.1. Study Period and Animals

This study analyzed data over four years from 2019 to 2022. This corresponds to the time when the fecal laboratory tests results were digitalized and when the changes to the parasite surveillance and management strategy were implemented.

All animals under managed care were included in the routine parasite surveillance program, except for aquatic animals and animals housed within large aviary-type settings because of the difficulties in obtaining diagnostic samples for laboratory analysis. The animals were managed in sections, which were generally taxa-based, although several sections were named after exhibit themes. The section names were only for organizational purposes and do not reflect the taxonomic classification of the individual animals within the section.

### 2.2. Sample Submission

Fecal samples from individual animals were the preferred sample for laboratory testing, but as this was logistically challenging to obtain due to the housing conditions and natural behavior of some animals, pooled fecal samples (aggregated fecal samples from multiple fecal piles in the enclosure) were also used for parasite surveillance. Pooled samples were usually from groups with no more than ten individuals. Fresh fecal samples were collected by the animal caretakers in labelled containers and submitted to the veterinary

hospital on the same day. The samples were kept chilled in a cooler box with ice packs or in a refrigerator at 4 °C until they were taken out for examination later in the day.

An information sheet at the sample drop-off point provided instructions on the accurate labelling of the samples with the identification number of the animal(s) and a link to an online sample submission form (Microsoft Forms; Microsoft Corporation, Redmon, WA, USA) that requested the following information:

- Date of sample collection
- Species
- Animal ID
- Purpose of test (routine/disease/recheck)

Tests labelled as "routine" were for healthy animals undergoing regular fecal examination. "Disease" tests were for parasite surveillance in animals displaying clinical signs of gastroenteritis during daily keeper checks, including weight loss, lethargy, and diarrhea, as part of a clinical diagnostic work-up requested by veterinarians. "Recheck" tests were post-treatment checks conducted approximately 2 weeks after the first treatment to assess the response to treatment.

### 2.3. Laboratory Analysis

Gastrointestinal parasites were diagnosed by means of both direct microscopic examination and saturated magnesium sulfate flotation methods. For direct microscopic examination, a small amount of feces (1–2 g) was emulsified in one or two drops of saline on a 1 mm thick microscope slide and examined after the placement of a coverslip under $40\times$ magnification. For saturated magnesium flotation, one spatula of feces (approximately 5 g) was suspended in a 15 mL conical tube and filled to the 15 mL mark with saturated magnesium sulfate solution (Kruuse Fasol, Jorgen Kruuse A/S, Langeskov, DK5550, Denmark). The tube was thereafter capped, then inverted a few times before being centrifuged for 5 min at 3500 rotations per minute. After centrifugation, the tube was topped up with the solution until a meniscus was formed. A coverslip was placed on the meniscus layer of the conical tube and left for 5 min, before being transferred to a 1 mm thick microscope slide and examined under $10\times$ magnification.

The presence of parasitic larvae or ova was reported by veterinary laboratory technicians in separate columns on the Excel sheet (Microsoft Excel; Microsoft Corporation, Redmon, WA, USA) generated by the online sample submission form. The identification of the parasites was performed via morphological analysis.

### 2.4. Medication

When gastrointestinal parasites were detected in tests and determined to be clinically relevant (generally non-commensal), a veterinarian dispensed the appropriate treatments. All animals receiving treatment were treated until the fecal test results were negative for parasites. In general, the presence of protozoa was insignificant unless accompanied by clinical signs like weight loss or diarrhea, which necessitated treatment with metronidazole. Coccidia was insignificant in adult animals but was treated with toltrazuril when detected in neonates only. Gastrointestinal metazoans were largely regarded as non-commensal and clinically relevant.

Gastrointestinal metazoans were treated with a larger range of anthelmintics, including benzimidazoles (fenbendazole and albendazole), pyrantel pamoate, macrocylic lactones (ivermectin), or salicylanilides (closantel). Veterinarians prescribed anthelmintics according to an anthelmintic rotation schedule involving fenbendazole, pyrantel pamoate, and ivermectin. The rotation schedule specified which drug should be the first, second, and third choice of anthelmintic for the year. The priority of anthelmintics changed the following year, with the second choice becoming the first, and the third choice becoming the second choice. Salicylanilides were used only in cases where all three drugs failed to be effective, which was noted only for *Haemonchus contortus* infections in the Turkmenian markhor (*Capra falconeri*).

*2.5. Test Schedule*

In the first year, all animals were screened for parasites twice a year, according to the previous test protocol. Subsequently, the test results were analyzed at the end of each calendar year. The total number of fecal laboratory analyses performed and the total number of times that treatments were dispensed were tabulated by section. The frequency of treatments (total number of times that treatments were dispensed/total number of tests performed) determined the fecal gastrointestinal parasite surveillance schedule for the following year:

- Sections with an anthelmintic treatment frequency under 10% would be tested only once the following year.
- Sections with a treatment frequency between 10% to 30% would be tested twice the following year.
- Sections with a treatment frequency above 30% would be tested four times the following year.

The schematic of the process of fecal testing for gastrointestinal parasite surveillance is summarized in Figure 1.

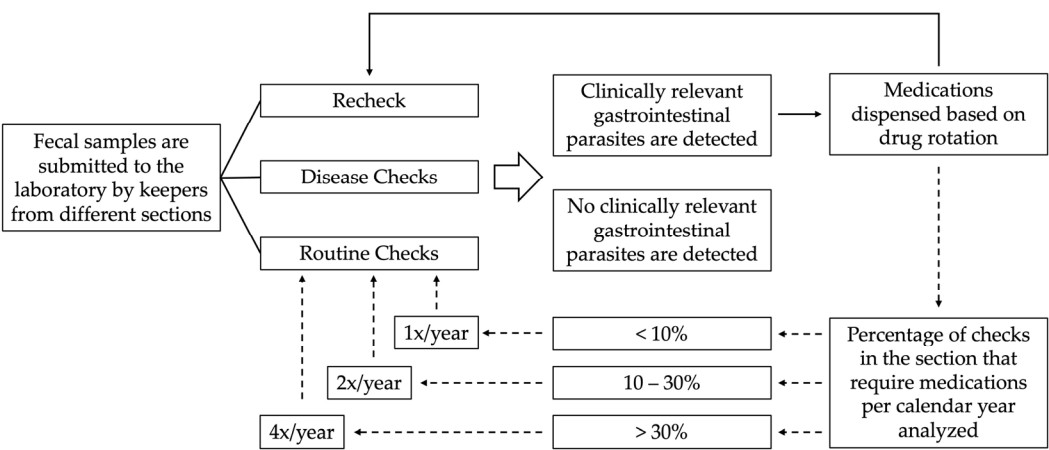

**Figure 1.** Schematic of fecal testing for gastrointestinal parasites and process which determines the following year's routine test frequency.

*2.6. Indicators of Health*

The total number of tests labelled "routine" and "diseased" was calculated each year. The number of times that medications were dispensed by the veterinarians was also calculated for these two groups of tests. Two parameters were used as indicators of the parasitic prevalence and health of the collection:

1. Percentage of "routine" tests requiring treatment with anthelmintics
2. Percentage of "disease" tests requiring treatment with anthelmintics

A higher percentage of routine tests requiring treatment meant that more animals had parasitic prevalences that were significant during routine tests and required medications. This was taken as an indicator of a higher general parasitic prevalence in the collection.

A higher percentage of disease tests requiring treatment meant that more animals were detected with clinically significant parasitic prevalence outside of the routine testing schedule. This was taken as an indicator that more animals had parasitic disease and therefore poorer health.

*2.7. Monitoring Anthelmintic Efficacy*

To monitor anthelmintic efficacy (ability of the anthelmintic to treat a parasite), the management of chronic *Strongyloides* spp. infection in orangutans (*Pongo* spp.) was compared between Year 1 and Year 4 of the study, which was when the same anthelmintic (fenbendazole) was the first-choice drug for the year. The total number of routine tests

and the number of times the animals were still positive for the parasite on the recheck test after the treatment course were tabulated to indicate the potential anthelmintic efficacy of the parasites.

*2.8. Statistical Analysis*

Chi-square tests were used to assess the statistical significance of the reduction in the total number of clinically significant tests between Year 1 and Year 4, as well as the reduction in the number of tests requiring treatment and the reduction in the number of recheck tests that were persistently positive for the parasite after one course of treatment between Years 1 and 4 in the zoo managed population of orangutans.

## 3. Results

The frequency of tests and treatments is summarized in Table 1. Between the first and the final year of the study, the total number of parasite tests performed reduced from 2126 to 1489 ($-30.0\%$). The number of animals with clinically significant parasite tests (requiring treatment) reduced from 451 to 244 ($-46.9\%$). A chi-square test assessed that the reduction in the number clinically significant tests between Year 1 and Year 4 was statistically significant ($p$ value $< 0.01$). Fewer animals presented as diseased with clinical parasitism.

**Table 1.** Frequency of fecal parasite tests performed and the number/percentage of clinically relevant (treatments prescribed) tests.

| Year | Routine | | | Disease | | |
|---|---|---|---|---|---|---|
| | Total Number of Tests | Tests Requiring Treatment | Percentage (%) | Total Number of Tests | Tests Requiring Treatment | Percentage (%) |
| 2019 | 1767 | 334 | 18.9 | 359 | 117 | 32.6 |
| 2020 | 1406 | 274 | 19.5 | 292 | 70 | 24.0 |
| 2021 | 1449 | 205 | 14.2 | 217 | 41 | 18.9 |
| 2022 | 1304 | 206 | 15.8 | 185 | 38 | 20.5 |

An overview of the distribution of the tests performed, along with the common gastrointestinal parasites detected, is provided in Table 2.

**Table 2.** Overview of the distribution of tests and common gastrointestinal parasites detected.

| Group of Animals | Percentage of Tests (%) | Common Parasites | |
|---|---|---|---|
| | | Animal | Parasites Detected |
| Avian | 10 | Columbiformes | Ascarids, Coccidia |
| | | Psittaciformes | Ascarids |
| | | Bucerotiformes | Ascarids, Coccidia. Strongyles, Strongyloides, |
| | | Phoenicopteriformes | Pinworms, Strongyles, Strongyloides |
| | | Piciformes | Capillaria |
| | | Galliformes | Coccidia |
| | | Casuariiformes | Flagellates |
| | | Strigiformes | Flagellates |

**Table 2.** *Cont.*

| Group of Animals | Percentage of Tests (%) | Common Parasites | |
|---|---|---|---|
| | | **Animal** | **Parasites Detected** |
| Reptile | 30 | Turtles | Flagellates, *Nyctotherus* spp., Strongyles, |
| | | Lizards | Ameba, Coccidia, Flagellates, Pinworms, Strongyloides, Strongyles |
| | | Snakes | Ascarids, Flagellates, Strongyles |
| | | Frogs | Ameba, Flagellates, Strongyloides, Strongyles |
| Carnivores | 15 | Viverrids | Flagellates, Strongyles, Strongyloides |
| | | Mustelids | Flagellates, Pinworm |
| | | Felids | Flagellates, Strongyles, Strongyloides, *Toxocara cati* |
| | | Ursids | Flagellates |
| | | Herpestids | Ascarids |
| Primate | 25 | Macaques (*Macaca* spp.) | Flagellates, Strongyles |
| | | Slow Loris (*Nycticebus* spp.) | Strongyles, Strongyloides |
| | | Great Apes | Strongyles, Strongyloides, Trichuris, *Troglodytella* spp. |
| | | Gibbons (Hylobatids) | Flagellates, Strongyloides |
| | | Colobine Monkeys | Strongyloides, Trichuris |
| | | Lemurs | Flagellates |
| Hoofstock * | 10 | Tapir | Strongyles, Strongyloides |
| | | Cervids | Strongyloides |
| | | Suids | Coccidia |
| | | Turkmenian Markhor (*Capra falconeri*) | Trichostrongyles—*Haemonchus contortus* |
| | | Equids | Strongyloides |
| | | Giraffe | Trichuris |
| | | Hippo | Coccidia |
| Other Mammals | 10 | Rodents | *Hymenolepis* spp., Strongyles, Strongyloides, Trichuris |
| | | Bats | Flagellates, Strongyles, Strongyloides |
| | | Xenarthra | Coccidia, Flagellates, Strongyloides, |
| | | Kangaroos | Labiostrongyles, Strongyles, Strongyloides |

\* Flagellates are a common finding in all hoofstock.

A comparison between the frequency of fecal parasite tests and the treatment of chronic strongyloidiasis in the collection of orangutans is summarized in Table 3. Between Years 1 and 4, the total number of tests performed reduced from 105 to 69 (−34.3%). A chi-square test proved that out of all the tests performed in each year, the reduction in the number of tests requiring treatment and the reduction in the number of recheck tests that were persistently positive for the parasite after one course of treatment between Years 1 and 4 were statistically significant (*p* value < 0.01).

**Table 3.** Comparison of the response of *Strongyloides* sp. in orangutans with fenbendazole between two years.

| Parameter | Year 1 (n = 23) | Year 4 (n = 20) |
|---|---|---|
| Total number of tests | 105 | 69 |
| Number of tests requiring treatment | 53 (53/105; 50.5%) | 11 (11/69; 15.9%) |
| Number of recheck tests that are still positive for parasites | 24 (24/53; 45.3%) | 2 (2/11; 18.1%) |

## 4. Discussion

The parasite management strategy in this study was able to reduce the total number of routine parasite tests performed annually without any negative effects on the parasitic prevalence and health of the animals in the collection. Between the first and last year of the study (2019 and 2022), 30.0% fewer fecal tests were performed, representing a large reduction in the amount of time required for laboratory technicians on routine surveillance. More importantly, although the number of tests performed was reduced, there was no compromise in the preventive healthcare of the animals in the collection, as evidenced by the general reduction in the percentage of routine tests requiring medical treatment, as well as the reduction in the percentage of animals presenting with clinical disease due to parasitism.

The increase in the frequency of parasite surveillance checks for sections with a higher frequency of treatment allows for the surveillance program to be targeted toward the animals that have a higher incidence of parasitism. Likewise, reducing the frequency of parasite surveillance checks for sections with a low frequency of treatment allocates these resources to the animal sections that require closer monitoring. It is possible that the closer surveillance of the animals with a higher incidence of parasitism was able to more proactively manage parasitic prevalence in these animals, thereby reducing the number of clinical cases of parasitism in these animals.

There are, however, some inherent problems with assessing the risk of parasitism in sections rather than individual animals. In a large group, individual animals with a high incidence of parasitism may be overlooked, especially if the rest of the animals within the section have reasonably low levels of parasitism, as overall the section may not reach the threshold required for a higher frequency of routine parasite checks in this protocol. Less importantly, animals with a high incidence of parasitism in sections with fewer animals may disproportionately increase the overall incidence rate, resulting in more frequent parasite checks in all of the animals in the section, even though they may not require a more intensive degree of surveillance. One way to better calibrate this is to identify these trends in the specific groups of animals within the section, increasing the frequency of parasite surveillance only for the groups of animals that have a higher incidence of clinically relevant parasitism. Similarly, pooled fecal samples in group-managed animals may overlook individuals with more significant levels of parasitism, although as treatment is administered to all animals if any gastrointestinal parasite is detected, these individuals are still likely to receive the required therapy.

The retrospective nature of this study indelibly comes with multiple factors that are not controllable. The number of tests performed annually, for instance, can be affected by the degree of compliance by the animal caretakers in submitting all the fecal samples required according to the schedule and the total number of animals in the collection. Whilst there will be inevitable variation in these factors, the assumption in this study is that they remain equally variable over the four years of the study and should not affect the test results obtained significantly. Another factor that can be variable is the general health status of the animals in the collection. The occurrence of an outbreak in a population of animals can result in an unusually high number of animals presenting with clinical disease for the year. External stress factors such as aberrant weather conditions or construction works nearby can also result in a greater incidence of parasitic disease [7]. These factors

remained reasonably constant over the years of the study and are unlikely to have affected the results obtained.

The key factor in determining whether the tests conducted yielded clinically relevant parasitic prevalence in this study is the frequency of treatment. This was used rather than the strict incidence of parasitism (fecal samples with parasites vs. fecal samples without parasites) because not every detection of a parasite is of clinical relevance, as some species generally classified as parasites could be commensal to the specific host species. The frequency of treatment is highly dependent on whether a veterinarian decides to dispense an anthelmintic based on the fecal test results. The individual variation regarding this clinical decision is minimized with the adherence to the treatment protocol as stipulated in this study, which was agreed upon by all veterinarians. The same team of veterinarians also managed the preventative healthcare and parasite control of the collection during the period of the study, minimizing further variation in this approach to parasite control.

A criticism of the parasite control strategy is the zero-tolerance approach to parasites, in that all animals are treated until no parasite larvae or ova were seen in the fecal examinations. This approach was taken because historically, animals have died of diseases directly related to parasitism despite having been seen with low burdens of parasites. The stance of administering treatment until being negative for parasites is perhaps one that errs on the side of caution, as parasitic disease seems to be able to develop very quickly. It also appears to be logical that if resources are allocated to conduct routine fecal surveillance tests for parasites, the animals that test positive should be the target for treatment. Such an intensive, risk-averse treatment approach, however, may eliminate the refugium population, which refers to the untreated hosts or environment that allow the maintenance of drug-sensitive parasites in the face of exposure [8]. This, in turn, may increasingly select for resistance alleles within the parasite population.

Many refugia-based parasite control strategies employed in the management of domestic livestock involve treating only a proportion of the group of animals, and most animals and worms remain unexposed to the drug, thereby conserving susceptible alleles [8]. The decision of which animals to treat has largely been based on fecal egg counts, animal production figures or even the color of mucus membranes [9–11]. In the setting of a zoological institute, however, many of these indicators will be challenging to assess, and are subjective and not a sensitive strategy to base the decision of deworming on. The physical examination of many of the animals will need to be performed under anesthesia, which is impractical for a large collection of animals. Wild animals under human care can mask many clinical signs of disease, making it difficult to identify individuals that may be clinically diseased until later in the disease progression. Performing quantitative fecal examinations, like fecal egg counts, as part of routine parasite surveillance for a large collection is also impractical as it requires greater manpower and resources to carry out. It is therefore reasonable that a positive fecal test is used as an indicator for anthelmintic treatment as it is in the described control strategy.

Another limitation of this control strategy is that every gastrointestinal parasite for every species of animal in the collection is treated the same way. This does not discern some parasite species (e.g., strongyles in ruminants) that are known to be well tolerated at low burdens, which could potentially be regarded as a commensal pathogen, thereby avoiding treatment and reducing the pressure for resistance selection. The parasite surveillance method for these animals could focus on more quantitative methods of parasite detection (e.g., McMaster's test) to reflect the parasitic burden and assist with clinical decisions on whether treatments are required or not, rather than employing the same qualitative approach to all parasites. Zoological institutions like MWG house many animals of high intrinsic value, which is the main justification for a strict zero-tolerance approach to gastrointestinal parasite management. With further data collection and the study of the incidence of parasitic disease in the wide range of species in the collection, perhaps parasite detection and treatment protocols can be altered, considering the true risk of debilitating parasitic diseases in the specific groups of animals. Other methods of parasite testing may

also be more sensitive methods for identifying particular parasites, like the Baermann technique to diagnose parasitic larvae or sedimentation tests to identify heavier parasite eggs (e.g., trematodes), although the addition of these tests will indelibly increase the time and manpower required to perform these routine tests.

An interesting finding in this study is regarding the increased anthelmintic efficacy of fenbendazole in treating *Strongyloides* sp. affecting the population of orangutans. Both the number of tests performed that necessitated treatment and the number of recheck tests that were still positive for the *Strongyloides* sp. after one course of treatment were reduced with fenbendazole, suggesting that the parasites were more susceptible to this drug after one full rotation of the drug classes in this control strategy. This finding is consistent with reports on the mechanisms of anthelmintic resistance and slowing the accumulation of resistance alleles in the parasites [12]. The management of the orangutans in the zoo has been the same in recent years, with a population that is stable and husbandry practices that have not been changed. Like many other non-human primates under human care, parasitic infections with *Strongyloides* spp. have been a perennial problem, as the parasite is capable of auto-infection and therefore maintains a constant level of infection in these animals [13,14]. Although not observed at MWG, *Strongyloides* spp. is capable of disseminated infections in orangutans, resulting in fatal disease [15]. Prior to the study, poor anthelmintic efficacy was observed in the management of strongyloidiasis in this population, with both fenbendazole and ivermectin being used with limited effect on the prevalence of the parasite. The poor response of *Strongyloides* spp. to fenbendazole was demonstrated with the high number of recheck tests that were still positive for the parasite even after one course of treatment. This is a stark contrast with the very low number of recheck tests that remained positive when fenbendazole was used again as the first-choice anthelmintic after three years. With the only relevant and perceptible difference being the adherence of the anthelmintic rotation protocol by all veterinarians, it is highly probable that the greater response of the *Strongyloides* spp. infections to the use of fenbendazole was due to this rotation of drugs.

The alternation of anthelmintics is generally practiced in farm animal medicine and is thought to select less intensively for resistance in a population of parasites [16,17]. The use of levamisole has been experimentally proven to select against benzimidazole resistance [18,19]. It is possible that the addition of pyrantel to the anthelmintic rotation schedule in this strategy further achieves greater selection against resistance to each individual drug. There are, however, other methods that may be superior to anthelmintic rotation in reducing drug resistance. A simulation model that investigated the optimal strategy for the use of multiple drug classes showed that drug resistance developed more slowly when a combination of two drugs was used compared with a rotation schedule [20]. The use of a combination of anthelmintics to manage resistant parasites was a common practice before the rotation schedule, with the choice of drugs largely based on the preferences of the prescribing veterinarian. Although no formal study was performed, several parasites affecting the animals appeared to be resistant to multiple drugs, as despite combination therapies, the fecal tests from certain animals were persistently positive. Anthelmintic combinations therefore may not be more effective in such a situation where the parasites already possess multiple resistance genes.

Many zoos adopt parasite control strategies that are based on the natural history of the various gastrointestinal parasites in temperate regions, where seasons have a significant impact on parasitic load, with higher loads expected in the warmer periods of the year [21]. Adopting these season-based parasite control strategies therefore will be limited in areas where the environmental temperature is high throughout the year. This risk-based parasite surveillance and management strategy therefore presents an alternative method to carry out a targeted parasite control program in areas where the temperature is constantly high throughout the year. There may be other meteorological conditions, like rainfall, that could affect the risk of parasite infections in the tropics [22]. Although no clear correlation between rainfall and parasitic infection was noted in this study, this can be a subject for

further research. If there is truly a correlation, perhaps parasite surveillance and treatment regimens can be adapted further to target higher parasitic activity during periods of higher precipitation.

### 5. Conclusions

The risk-based gastrointestinal parasite control strategy represents a targeted method of managing parasitic infection in wild animals under human care in the tropics. This targeted approach, allied with an annual anthelmintic drug class rotation, reduces the number of routine tests that are performed annually for the surveillance of parasites in large collections without compromising the health of the animals, achieving the aims of preventative healthcare in a zoological collection.

**Author Contributions:** Conceptualization, Y.H.; methodology, Y.H.; validation, Y.H. and D.H.H.C.; formal analysis, Y.H.; investigation Y.H. and D.H.H.C.; resources Y.H.; data curation, Y.H.; writing-original draft preparation, Y.H. and D.H.H.C.; writing-review and editing, Y.H.; visualization, Y.H.; supervision, Y.H.; project administration, Y.H. All authors have read and agreed to the published version of the manuscript.

**Funding:** This research received no external funding.

**Institutional Review Board Statement:** Ethical review and approval were waived for this study due to the retrospective nature of the study, analyzing work that has already been done without consideration for the study.

**Data Availability Statement:** Data is contained within the article.

**Acknowledgments:** The authors would like to thank all veterinary and animal care staff at Mandai Wildlife Group for their enthusiasm in the preventive healthcare of the animals.

**Conflicts of Interest:** The authors declare no conflicts of interest.

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
