# Peer review of "Risk-Based Gastrointestinal Parasite Control in a Tropical Zoological Institute"

_2673-5636, doi:10.3390/jzbg5020014_

Round 1

Reviewer 1 Report

Comments and Suggestions for Authors

The authors improved the results and adapted the discussion.

Author Response

Dear Reviewer 1

Thank you for taking the time to review this manuscript again, and your encouraging comments! I've attached my responses to your comments, along with the other reviewers' comments in the event that you would like to view all the changes that have been made to this manuscript after this round of reviews. 

Thank you!

Reviewer 2 Report

Comments and Suggestions for Authors

Dear authors, congratulations on your highly interesting manuscript. This targeted approach as part of a preventative medical program in animals kept under professional care shows a clear hands-on understanding of zoo animal disease management. I would like to point out some suggestions and ask some specific questions:

-    Abstract – lines 9 and 10 – I would suggest some term modification, as “gastrointestinal parasites are seen in higher numbers” may be of subjective interpretation. You may use, for example, “(…) where the burden of gastrointestinal parasites on fecal testing is high.”;

-      Introduction – line 38 – The influence of the environment can be a stressful factor per se, as you mention in the discussion section, so I would suggest mentioning one or two examples of zoo stress factors (including, if you may, weather conditions);

-        Introduction – line 57 - I would add some ground and succinct information on the Mandai Wildlife Group, briefly including the groups of animals included in their collection. The reader only learns in what groups of animals was this parasite control applied in the Results section;

-    Discussion – initiating at line 238 – When evaluating the problems with the assessment of individual samples versus fecal pools, it would be interesting to mention the dimension of the bigger groups of animals in your institution;

-  Discussion – line 290 – Please add that, these indicators are indeed challenging to assess in a zoo setting, but they are, in fact, subjective and not a sensitive strategy to base the decision of deworming;

-     In each fecal evaluation, was the collection of samples done only once per individual/per area? To increase the sensitivity of the fecal examination, it would be interesting to examine serial samples (i.e. collected in three alternated days), as this would not increase the work burden laboratory-wise.

Author Response

Dear Reviewer 2

Thank you for taking the time to review this manuscript again, and your encouraging comments! I've attached my responses to your comments, along with the other reviewers' comments in the event that you would like to view all the changes that have been made to this manuscript after this round of reviews. 

Thank you!

Reviewer 3 Report

Comments and Suggestions for Authors

The present study examines a risk-based parasite management strategy implemented in a tropical zoological institute. By utilizing risk-based routine fecal tests and a yearly drug-class rotation, the study aims to reduce the frequency of parasitic disease in the animal collection while restructuring preventative healthcare efforts. The findings suggest a decrease in the number of animals with clinically significant parasitic disease as well as a reduction in the number of fecal tests performed over a four-year period, highlighting that this targeted approach could prove to be effective.

I believe the present study elucidates an innovative strategy that could prove effective, addressing concerns such as anthelmintic resistance and resource management, which are crucial considerations for implementation in zoological institutes worldwide. Despite that, I have significant concerns regarding the methodology employed in the study and the conclusions drawn in its current state.

The abstract effectively communicates the key aspects of the study but would benefit from grammar revisions and minor adjustments to enhance clarity and specificity. For instance, at line 7, the sentence should read: The surveillance and treatment of parasites ARE.... Similarly, at line 18, the sentence should be revised to: 'Anthelmintic class rotation also improved the efficacy of fenbendazole IN.... Furthermore, adding specific details regarding the methods used to determine parasite prevalence and providing quantitative or percentage reductions of parasite tests and clinically significant parasitic disease cases should be added to strengthen the abstract's impact and provide a clearer understanding of the study's outcomes.

Regarding the methods and discussion, it is not clear whether whenever an animal presents clinical signs of parasitism, it is tested regardless of its parasitic record in the previous year, or if this "disease check" classification is only done during planned testing. Also, it is important to clarify the frequency at which animals are evaluated to understand if they present clinical signs of parasitism.

The present study does not use methods that concentrate heavier parasite eggs such as trematode eggs, which in this study could only be diagnosed in direct smears, what could significantly reduce their diagnosis. I recommend that in future work and in the zoo's own tests, they include sedimentation as an analysis method, analyzing the sediment resulting from flotation with a drop of methylene blue to more easily distinguish viable parasite eggs from residues. They can also add the Baermann technique to diagnose parasitic larval forms. I am not of the opinion that the non-performance of these methods invalidates the importance of this study; however, they must dedicate some sentences in the discussion to explain this limitation of the study and that it may underdiagnose trematodes and larval forms.

Table 3 should have relative results in percentage, like Table 1: Percentage of tests requiring treatment and the percentage of recheck tests positive for parasites.

Furthermore, in the present study the authors compare the 4 years in which the new selective testing methods have already been implemented, not comparing results with any year in which this had not yet been implemented. It is very important to compare the results with those of the year 2018, before implementing both selective testing and antiparasitic rotation, to know if it was really a significant difference. If unable to compare with the results of the year 2018, due to lack of computerization of the data, it would again be very important to explain this limitation in the discussion and to be very careful with what can or cannot be concluded with the study as it stands.

In Line 125 should read: centrifuged.

The sentence ending in line 267 should be changed because "parasites may be commensal to the species" is contradictory. It should be changed to something like: species generally classified as parasites could be commensal to the specific host species.

No statistical test was conducted to assess whether the decrease in the number of tests, the number of tests requiring treatment or even the response of Strongyloides sp. in orangutans with fenbendazole are statistically significant with the change in management. Consequently, nothing definitive can be concluded from the current state of the study, as these observed changes could potentially be attributed to chance without a p-value test lower than 0.05. Unlike other methodological issues in this study, the absence of proper statistical analysis cannot be justified solely in the discussion; rather, such analysis should have been an essential component. However, I remain confident that the results are robust enough to yield statistically significant findings when properly tested and that stronger study can be published after that.

Regarding the conclusion, the authors simultaneously adopted the change in the frequency of tests based on the need for treatment of these animals in the previous year and the implementation of an anthelmintic rotation each year, which makes it not possible to conclude which of these changes was responsible for the decrease in tests needed and in tests requiring treatment (counting that an animal after treatment is tested and retreated until the result is negative, resulting in a decrease in tests needed and tests that require treatment when the medication is more effective due to a drug-class rotation).

The authors cannot conclude that this method is better than the previous one because they do not present the results from the year prior to the implementation of this new method; they can only conclude that the new method improves from year to year. Even then, they cannot conclude this without using some statistical test and having a p-value less than 0.05.

After statistically analyzing your results and if they are statistically significant, you can conclude:

This targeted approach, ALLIED WITH A YEARLY DRUG-CLASS ROTATION, reduces the number of routine tests ANNUALLY that are performed for the surveillance of parasites in large collections without compromise to the health of the animals, achieving the aims of preventative healthcare in a zoological collection.

Comments on the Quality of English Language

Some minor revisions are needed to improve the quality of English, but I have already provided my suggestions on what should be changed in the text above.

Author Response

Dear Reviewer 3

Thank you for taking the time to review this manuscript again, and your encouraging comments! I've attached my responses to your comments, along with the other reviewers' comments in the event that you would like to view all the changes that have been made to this manuscript after this round of reviews. 

Thank you!

Reviewer 4 Report

Comments and Suggestions for Authors

Summary

The authors propose a strategy of parasite management in a tropical zoological institute, based in the results of routine faecal checks and the appearance of clinically relevant parasitism, in order to determine the number of tests for next year. They develop a rotational protocol for anthelmintic treatment. They present some data of anthelmintic resistance, bur only for one parasite.

General concept comments

- Review introduction with more clear concepts related to “clinically parasitism” or “clinically relevant gastrointestinal parasites”.

- Material and methods can be improve with any basic data of animals studied, sample size (number of animals in the institute), laboratory analysis (quantity of faecal samples), medication prescription and anthelmintic efficacy.

- Results can be improved with more data of animals tested/no tested.

A better explanation of table 1 is recommended.

- Discussion and conclusions should include references and criterium to “clinically relevant parasitic prevalence”, and anthelmintic efficacy.

It is an interesting work about the management of parasites in zoological institutes.

Specific comments

Introduction

The main question is the concept of “clinically relevant gastrointestinal parasites”.  It can be different for different parasites and circumstances. When do you consider a parasite as “relevant”?

Material and methods

Line 82. Point 2.1. Study period and animals.

A brief reference should be included about animals studied. There is more information on supplementary material, but you can include any data on it.

Line 97. Point 2.2.

“pooled fecal samples”… How many animals can be tested in each pool?

Line 111:

“regular fecal examination”… What do you mean “regular”? Which is the protocol for testing the animals?

Line 122:

“one spatula of faeces”? Please explain the quantity of faeces in the “spatula”.

Line 136:

When do you consider a parasite “clinically relevant”?

Line 138:

“All animals are treated until…” What do you mean? All the animals of the section, all the animals positive, all the animals with clinical signs?

Line 182:

How do you asses the “parasitic prevalences that were significant”? Is there any reference?

Results

Line 204:

Table 1 needs more explanation.

The “tests requiring treatment” are all the animals with parasites? Or they are only the clinically affected animals?

Discussion

Just the same comments that before: can you improve the concept of clinically relevant parasitism? It can be different in any place.

Author Response

Dear Reviewer 4

Thank you for taking the time to review this manuscript again, and your encouraging comments! I've attached my responses to your comments, along with the other reviewers' comments in the event that you would like to view all the changes that have been made to this manuscript after this round of reviews. 

Thank you!

Round 2

Reviewer 3 Report

Comments and Suggestions for Authors

This revised version, which incorporates all the requested information and changes, represents a significantly stronger paper. However, there are a few minor adjustments I would recommend:

The authors should include the requested statistical analysis in the Materials and Methods section. I propose adding it as a separate point: 2.8 Statistical Analysis; Chi-square tests were used to assess…

In line 14 should read: flotations

In lines 211 and 229, it is preferable to write: p value<0.01

Author Response

Thank you for taking the time to review the manuscript. My responses are attached in the word document!
